# Multi-Objective Optimization of Process Parameters in 6016 Aluminum Alloy Hot Stamping Using Taguchi-Grey Relational Analysis

**DOI:** 10.3390/ma15238350

**Published:** 2022-11-24

**Authors:** Binghe Jiang, Jianghua Huang, Hongping Ma, Huijun Zhao, Hongchao Ji

**Affiliations:** 1School of Mechanical and Energy Engineering, Zhejiang University of Science and Technology, Hangzhou 310023, China; 2School of Mechanical Engineering, Hangzhou Dianzi University, Hangzhou 310018, China; 3School of Mechanical Engineering, North China University of Science and Technology, Tangshan 063210, China

**Keywords:** aluminum alloy, hot stamping, process parameters, grey relation analysis, orthogonal design method

## Abstract

The hot stamping technology of aluminum alloy is of great significance for realizing the light weight of the automobile body, and the proper process parameters are important conditions to obtain excellent aluminum alloy parts. In this paper, the thermal deformation behavior of 6016 aluminum alloy at a high temperature is experimentally studied to provide a theoretical basis for a finite element model. With the help of blank stamping finite element software, a numerical model of a 6016 aluminum alloy automobile windshield beam during hot stamping was established. The finite element model was verified by a forming experiment. Then, the effect of the process parameters, including blank holder force, die gap, forming temperature, friction coefficient, and stamping speed on aluminum alloy formability were investigated using Taguchi design, grey relational analysis (GRA), and analysis of variance (ANOVA). Stamping tests were arranged at temperatures between 480 and 570 °C, blank holder force between 20 and 50 kN, stamping speed between 50 and 200 mm/s, die gap between 1.05 t and 1.20 t (t is the thickness of the sheet), and friction coefficient between 0.15 and 0.60. It was found that the significant factors affecting the forming quality of the hot-stamped parts were blank holder force and stamping speed, with influence significance of 28.64% and 34.09%, respectively. The optimal parameters for hot stamping of the automobile windshield beam by the above analysis are that the die gap is 1.05 t, the blank temperature is 540 °C, the coefficient of friction is 0.15, stamping speed is 200 mm/s, and blank holder force is 50 kN. The optimized maximum thickening rate is 4.87% and the maximum thinning rate is 9.00%. The optimization method used in this paper and the results of the process parameter optimization provide reference values for the optimization of hot stamping forming.

## 1. Introduction

In the wake of the rapid growth of industrial development, the environmental problem has become a significant issue concerning the development of countries all over the world [1,2]. The light weight of automobiles is one of the solutions to this problem. Advanced manufacturing processes and lightweight materials in automotive production are the main method to improve the light weight and safety [3]. Aluminum alloy is known for its high specific strength, strong corrosion resistance, and other excellent comprehensive performance. 6xxx aluminum alloy, including 6016 aluminum alloy as a member of the lightweight material, and heat treatable alloy are commonly used in the automobiles external body panels and structural parts [4]. However, the plasticity of aluminum alloy is low at room temperature and the forming is not easily controlled. For 6016 aluminum alloy, the elongation is 6% to 8.5% when forming without heating, making them prone to cracking during forming complex parts [5]. Hence, aluminum alloy hot stamping process was proposed to effectively improve the formability of aluminum alloy while maintaining the mechanical strength [6,7].

In recent years, many attempts have been made to research the formability in hot stamping of aluminum alloys. Fakir et al. [8] proposed a finite element model to analysis the effect of forming temperature and stamping speed on the formability of AA5754, and the results showed that a higher forming speed could improve the formability of AA5754. Liu et al. [9] studied the effect of blank holder force on the formability of an AA7075 B-pillar reinforced panel; they found that forming defects could be improved by a reasonable blank holder force during hot stamping. Ma et al. [10] investigated the influences of friction coefficient on the formability of AA6111 and determined the optimal friction coefficient to be 0.15. Ma et al. [11] quantitatively analyzed the effects of the forming temperature, stamping speed, friction coefficient, and blank holder force on the formability on AA6082 during hot stamping, and found that the main factors affecting the formability of hot stamped parts are blank holder force and friction coefficient. For 6016 aluminum alloy, Kumar et al. [12] investigated the influences of the solutionizing, heating rate, pre-deformation, and cooling rate on the part mechanical properties of the AW-6016 alloy during simultaneous hot stamping and quenching and found that increasing the cooling rate after solution heat treatment had a positive effect on the part properties. The effect of initial blank temperature, blank holder force, die closing pressure, and die corner radius on the springback of AA6016 during hot stamping were studied by Ma et al. [5]. It is found that springback decreases significantly as the initial blank temperature rises up to 500 °C. To the best knowledge of the authors, there is little literature on the research of the formability of AA6016 during the hot stamping process. Based on above discussions, process parameters are vital for hot stamped forming, and determining the appropriate process parameters is a primary goal in the forming of hot stamping.

Zhou et al. [13] explored the influences of blank hold force and stamping speed on AA6111 forming quality and used the Latin hypercube method and NSGA-II to optimize the process parameters. Xiao et al. [14] analyzed the effects of stochastic variability of parameters such as blank holder force, friction coefficient, stamping speed, and blank temperature on forming quality during hot stamping, using the Monte Carlo simulation (MCS), response surface methodology (RSM) with NSGA-II to optimize the parameters. Xiao et al. [15] applied partition temperature control to study the partition temperature on the aluminum alloys’ forming quality and using the response surface methodology (RSM) with NSGA-II to optimize the parameters. In addition, Xie et al. [16] proposed a new mixed surrogate model based on PBM-BPNN and improved the multi-objective particle swarm optimization (MOPSO) algorithm to optimize the process on the thickness of double-C during stamping. However, the multi-objective optimization of the above process parameters was performed by building surrogate models with an intelligent algorithm, and the model accuracy is dependent on the sample size and the regularity of their results. Improving the accuracy of surrogate models has significantly increased the computational and experimental costs for researchers. This method could provide a range of non-dominant optimal solutions, but it still requires additional decision-making to choose a compromise solution, which adds more costs.

Grey relational analysis (GRA) combined with the Taguchi method can solve multi-response problems, and the method has been widely used in various fields. For example, Rekha R et al. [17] optimized the surface quality in austenitic stainless steel 304 during cylindrical grinding and analyzed the effects of the process parameters on surface roughness and material removal rate by Taguchi and GRA. Boukraa M et al. [18] used Taguchi with GRA to identify the optimal level of the friction stir welding process parameters for AA2195-T8. To the best of our knowledge, the Taguchi design with grey relational analysis (GRA) has not been used in the field of hot stamping.

In this study, hot tensile tests were used to evaluate the high-temperature mechanical properties of 6016 aluminum alloys. In addition, a finite element model of the automobile windshield beam was established. The Taguchi method was used to conduct orthogonal experiments, varying the levels of the die gap, sheet temperature, blank holder force, stamping speed, and friction coefficient for the experiments. Using grey relational analysis in statistical analysis, the multi-objective optimization question was transformed into a single-objective optimization question, and the contribution of each variable to the results was evaluated using variance analysis to optimize the maximum thinning rate and the maximum thickening rate of the part. The final combination of optimal levels was adopted and simulations were carried out. The results show that the forming performance of the component is significantly improved.

## 2. Materials and Experiment

### 2.1. Material

In the experiment, the cold-rolled 6016-T6 aluminum alloy with a thickness of 1.5 mm was selected. The chemical composition of the materials is listed in Table 1.

### 2.2. High Temperature Tensile Test

The structure and dimensions of the tensile specimen obtained by wire cutting are shown in Figure 1, with the length direction parallel to the rolling direction.

During the test, a thermocouple is welded to the middle surface of the specimen to test the temperature of the specimen. The heating time and cooling rate are controlled, and a 15-mm scale (15 mm for the mean temperature section) is set by scribing to correct the stress–strain curve and to obtain the true stress–strain curve by calculation. The test process is carried out on the Gleeble-3500, manufactured by FULETE Instrument Technology (Shanghai) Company Limited (Shanghai, China).

To make the experimental results accurately reflect the mechanical behavior of 6016 aluminum alloy during hot stamping, a high-temperature tensile experimental program for 6016 aluminum alloy was developed according to the range of the deformation temperature and strain rate of the hot stamping process; the temperature and the deformation experienced by the aluminum alloy specimen during the experiment is shown in Figure 2.

The specific step are as follows:

Step 1, the aluminum alloy specimen was first heated to 550 °C at a rate of 10 °C/s.

Step 2, the temperature was held for 3 min.

Step 3, the deformation temperature was lowered at a rate of 5 °C/s.

Step 4, the temperature was held for 2 min.

Step 5, the tensile test was carried out until the tensile fracture according to a specific deformation and strain rate, which was 400 °C, 450 °C, 500 °C, 550 °C, and 0.01 s^−1^, 0.1 s^−1^, 1 s^−1^, 10 s^−1^, respectively.

Step 6, immediately after deformation, the specimen was cooled rapidly by argon gas to preserve its high-temperature microstructure.

Figure 3 shows the true stress–strain curves of 6016 aluminum alloy at different deformation temperatures and different strain rates.

It can be seen that the thermal deformation of 6016 aluminum alloy is divided into three stages: strain hardening, steady-state deformation, and dynamic recrystallization softening, and there are both hardening and dynamic softening processes in the thermal deformation of 6016 aluminum alloy [19]. In the initial stages of deformation, the true stress rises rapidly with increasing strain, due to the increase in dislocation density and the interaction between dislocations increasing their resistance to movement. At the same time, due to the small strain, its intracrystalline storage energy is low and the dynamic softening process is difficult to carry out. As the strain increases, the dynamic softening and dynamic hardening tend to equilibrate. Later, under the action of the outer stress and thermal activation energy, reorganization or merges occur, resulting in dynamic recrystallisation.

As seen from Figure 3a, with the strain rate increases, the stress level also rises, indicating that 6016 aluminum alloy is a positive strain rate sensitive material. From Figure 3b, the peak stress obviously decrease as the temperature rises, which shows that the effect of temperature on peak stress is very significant. This is due to the fact that as the temperature increases, the thermal activation energy of the material increases, the critical shear stress decreases, the resistance to dislocation movement decreases, and the average kinetic energy and diffusion rate of the atoms of the metal increases, resulting in a reduction in stress. These discussion about the thermal deformation behavior of 6016 aluminum alloy at high temperature is consistent with the description by Huang et al. [20].

### 2.3. Finite Element Modeling of Automobile Windshield Beam

The elements in the finite element model are thermal–mechanical coupled. The Belytschko–Tsay shell cell is used to mesh the geometric model, which is realized by the automatic dissection function, where the sheet size is 1750 mm × 1420 mm, the minimum element size is 5 mm, and the whole sheet mesh is used in the calculation with 4 levels of adaptive optimization coefficients. The number of blank elements is 530, for punch is 31,398, and the die is 35,470. The stress–strain data obtained in Figure 4 were coupled into the finite element material model. The upper and lower die deformation is minimal, so it is set as a rigid body, and the thickness of the sheet is set to 1.5 mm. H13 is chosen as the material of the die. Table 2 demonstrates the thermal and physical properties of the 6016 aluminum alloy. The density of the specimen is 2.7 kg/m^3^ and the Poisson ratio is 0.3.

The temperature of the die is 10 °C and the temperature of the blank reaches about 500 °C. The temperature between the die and the sheet is huge, and a large amount of heat is exchanged between the sheet and the die at the contact interface. The interfacial heat exchange coefficient between the die and the sheet is related to the pressure and distance, as shown in Table 3.

When using one-step stamping and forming, excessive thinning and even cracking occurred in the middle raised part because of uneven size distribution. To improve this forming defect, the two-step forming method is used in this paper. The stamping process can be described as follows: firstly, the ejector is moved up to form the middle raised part, and after this part is formed, the punch and ejector press the middle raised part together, and the punch is lowered at a certain stamping speed until the end of the form. The established finite element model is shown in Figure 4.

### 2.4. Hot Stamping Forming Experiment

To verify the accuracy of the finite element model, this paper confirms the temperature change during the forming process of the sheet and the thickness distribution of the sheet after the forming is finished, respectively. The toolset used for hot stamping tests mainly consists of a pad plate, flow divider, punch core, die core, guide, positioning block, etc. The cooling pipes are connected in parallel to facilitate the installation and sealing of the toolset. To facilitate the processing of cooling pipes, the punch and die cores are divided into blocks. The 3D model is shown in Figure 5a, and the actual model of the mold is shown in Figure 5b.

The initial temperature of the sheet is set to 510 °C. By putting sensitive thermocouples in the sheet at the pre-buried position, the temperature data at this point are monitored in real time by the sensor when the hot stamping experiment is conducted. The experimental detail and experimental equipment are shown in Figure 6.

## 3. Results and Discussion

### 3.1. Simulation Analysis and Validation

The blank holder force is 20 kN, the friction coefficient is 0.15, the initial temperature of the sheet is 510 °C, the die gap is 1.05 t, and the stamping speed is 50 mm/s for the initial simulation. The thickness, stress, strain, temperature clouds are shown in Figure 7.

As seen from Figure 7a, the maximum thickening rate and maximum thinning rate were in the edge plane cross-angled area, and there is no change in thickness in other areas of the part. Under hot stamping conditions, the flow stress gradually decreases, and the forming limit gradually increases with the increasing temperature of the sheet. However, as the middle raised part formed over, the smaller the flow stress, the easier the material flows, making it easy for the material in the middle raised part to thin at the stress concentration when it flows into the edge plane cross-angled area due to the uneven material flow. Thus, there is a distribution of both maximum and maximum thickness in the edge plane cross-angled area. Figure 7b shows a stress concentration at the cross-angle area, so there is thinning in this area. Figure 7c shows that the maximum equivalent stress of the part is 0.174, which avoids the problem of cracking in the forming of components. Figure 7d shows that the temperature in the middle raised area is lower because the forming of the middle raised area is finished, and the middle raised area is in contact with the mold, which removes a lot of heat loss.

After the hot stamping experiment, the data from the sensor monitoring device were read to compare with the simulated simulation data, and the results are shown in Figure 8a. The cross-section was also cut using a wire cutter and the thickness of the cross-sections was measured using a vernier caliper. The results are shown in Figure 8b, and it can be seen that both the thickness distribution and temperature variation show good agreement with the experimental results, which verifies the reliability and accuracy of the finite element model developed in this paper.

### 3.2. Optimization of Hot Stamping Process Parameters

#### 3.2.1. Design Variables

Taguchi’s method is a robust design method that emphasizes design rather than testing [21,22,23]. The two experimental design tools used by Taguchi’s method are orthogonal experiments and signal-to-noise ratio. Orthogonal experiments are utilized to ascertain the combination of factors and their levels to reduce the number of experiments reasonably [24]. The signal-to-noise ratio is used to measure the proportion of a factor’s contribution to the results. The maximum thinning rate and maximum thickening rate in the forming result are used as the quality indicators of the formed parts [25]. The larger maximum thinning rate and the maximum thickening rate, the larger possibility of fracture and wrinkle defects, and they are calculated according to Equations (1) and (2).
(1)$$Thinning=\frac{initial\;blank\;thickness-\min imum\;part\;thickness}{initial\;blank\;thickness}\times 100\%$$
(2)$$Thickening=\frac{\max imum\;part\;thickness-initial\;blank\;thickness}{initial\;blank\;thickness}\times 100\%$$

In the expression of the signal-to-noise ratio, there are two characteristics, ‘the larger the better’ response as expressed in Equation (3), and ‘the smaller the better’ response as expressed in Equation (4). In this paper, we need to minimize the results and we choose Equation (4) to handle our data:(3)$$\eta_{Larger}=-10\mathrm{lg}\left(\frac{1}{n}\sum_{i=1}^{n}\frac{1}{y_{i}^{2}}\right)$$
(4)$$\eta_{Smaller}=-10\mathrm{lg}\left(\frac{1}{n}\sum_{i=1}^{n}y_{i}^{2}\right)$$
where η is the signal-to-noise ratio, $y_{i}$ is the $i^{th}$ response in all results, and n is the number of repetitions.

The finite element model is established to be used for studying the effects of the blank holder force, sheet temperature, die gap, stamping speed, and friction coefficient on the maximum thinning rate and maximum thickening rate after forming. Given the fact that too fast a stamping speed can easily cause product cracking and too slow cause much heat loss, the stamping speed was selected from 50 mm/s to 200 mm/s. As for the die gap, if the die gap is too large, the surface of the hot-stamped part will be easily wrinkled. If the die gap is too small, it will be easy to rub the workpiece and the die. So, the die gap is 1.05 to 1.20 times the thickness of the sheet, and the temperature is set at 480 °C to 570 °C for successful forming.

An orthogonal test with five factors and four levels was established. Table 4 shows the level table of the orthogonal experiment for finite element simulation.

When the factors and levels were determined, an orthogonal experiment with 5 factors and 4 levels was designed, and the L16(4^5^) orthogonal array was selected as shown in Table 5.

#### 3.2.2. Grey Relational Analysis (GRA) and Analysis of Variance (ANOVA)

Grey relational analysis (GRA), devised by researchers, is an efficient tool for solving inter-relationships among quality characteristics of multiple responses [26]. The GRA converts a multi-objective optimization question into a single-objective optimization question by converting the values of several objectives into the values of a single objective. Usually, the GRA method has three steps. In the first step, we need to produce the grey relationship with the help of the two characteristics mentioned above, depending on whether we need to minimize or maximize it, and then normalize it (between 0 and 1). In the second step, we need to produce the grey relational coefficient (GRC), the relationship between the multiple targets we establish and our factors. The third step is to obtain the grey relational grade (GRG), which is the average of the previously obtained GRC values.

Since there are different units and magnitudes of each factor, the obtained S/N ratio results need to be normalized. The normalized method is usually described as follows:

For the larger the better response:(5)$$X_{i}\left(k\right)=\frac{Y_{i}\left(k\right)-\min Y_{i}\left(k\right)}{\max Y_{i}\left(k\right)-\min Y_{i}\left(k\right)}$$

For the smaller the better response:(6)$$X_{i}\left(k\right)=\frac{\max Y_{i}\left(k\right)-Y_{i}\left(k\right)}{\max Y_{i}\left(k\right)-\min Y_{i}\left(k\right)}$$
where $X_{i}\left(k\right)$ and $Y_{i}\left(k\right)$ are the normalized and simulations values for the response k, respectively, $\max Y_{i}\left(k\right)$ is the maximum value for the k response in all simulations, $\min Y_{i}\left(k\right)$ is the minimum value for the k response in all simulations.
(7)$$\xi_{i}=\frac{\Delta_{\min}+\rho \;\Delta_{\max}}{\Delta_{oi}\left(k\right)+\rho \;\Delta_{\max}}$$
where $\xi_{i}$ is the correlation coefficient of the one-to-one correspondence between the comparison sequence and the reference sequence for the response k, $\Delta_{oi}\left(k\right)$ is the difference between absolute values for the response k, ρ is the grey relational distinguishing coefficient, generally, ρ is 0.5. $\Delta_{\min}$ is the lowest value in the $Y_{i}\left(k\right)$, $\Delta_{\max}$ is the highest value in the $Y_{i}\left(k\right)$.

Using the Taguchi method and grey correlation method above, the whole optimization process is shown in Figure 9.

Firstly, the signal-to-noise ratio analysis was performed using Equation (4) and the experimental results were normalized according to Equations (6) and (7) and scaled between 0 to 1. The results are shown in Table 6.

The above average values of grey relational grade (GRG) results for the different levels of the process parameters were shown in Table 7.

Due to the lack of degrees of freedom, we discarded the fifth-ranked die gap (B) and performed an ANOVA on the remaining process parameters to determine the percentage influence of each parameter on the results, and the ranked results are shown in Table 8.

From Table 7 and Table 8, it can be seen that the process parameters affecting the forming quality are: stamping speed > blank holder force > sheet temperature > friction coefficient > die gap, and the impact of stamping speed and blank holder force on the forming quality is 34.09% and 28.64%, respectively. Mean grey relational grade reveals that the higher the mean value, the greater the influence of that level on the results. To see it more clearly, we draw it out as shown in Figure 10. Then, we can get the best combination of process parameters as A4B1C4D1E4.

### 3.3. Optimization Results Validation

Using the optimized process parameters obtained above for simulation tests, the thickness distribution shown in Figure 11 is obtained, and the maximum thickening thickness rate is 4.87% and the maximum thinning rate is 9.00%.

The optimized maximum thickening rate performed much better than the result in the orthogonal test table, and the maximum thinning rate is in the optimal value area in the orthogonal test table. From Figure 11, the maximum thickening and maximum thinning are still concentrated in the edge plane cross-angled area, and the cracking in the middle raised part has also been optimized. To illustrate the reasonableness of optimization results for parameters. For friction coefficient, the optimization result is 0.15, which is in accordance with the conclusion of Ma et al. [10]. For the stamping speed, a good forming quality is conducted by 200 mm/s, which is also in accordance with the conclusion of Fakir et al. [8]. It can be seen that the sheet forming quality is improved to some extent by process parameters optimization, and it also shows that the grey relational analysis method and Taguchi orthogonal test are feasible in improving the sheet forming quality. In the case of this work, although the solution obtained by this method is not a Pareto solution set obtained by combing a surrogate model with intelligent algorithm, the solution obtained by statistical analysis approximates to a certain extent the solution in the Pareto optimal, which is a solution in ‘weak Pareto optimal’, while this method requires less of a sample size and regularity of the results, and can obtain a reliable solution in a relatively short time and with smaller computational effort.

## 4. Conclusions

In this study, the automobile windshield beam was considered as the object of study, and the maximum thickening and thinning rates of the sheet with different process parameters were obtained by the finite element model, and the finite element model was verified experimentally. The process parameters affecting the quality of the sheet forming were quantitatively analyzed by grey relational analysis (GRA) and analysis of variance (ANOVA). The main conclusions are as follows.Based on finite element software, a numerical simulation model of the 6016 aluminum alloy automobile windshield beam was established. At the same time, the hot stamping simulation experiment was conducted to compare the thickness distribution of the sheet after forming and the temperature variation during the forming process. The results showed that the thickness distribution and temperature variation of the simulation test were consistent with the trend of the experiment, which verified the reliability of the finite element simulation model.The number of tests was successfully reduced to an acceptable range by combining grey relational analysis with Taguchi’s orthogonal test, and the double-objective optimization problem was transformed into a single-objective optimization problem, and the importance of the factors influencing the forming was ranked: stamping speed, blank holder force, deformation temperature, coefficient of friction, and die gap. The impact of stamping speed and blank holder force on the formability is 34.09% and 28.64%, respectively.The optimization results indicated that the maximum thickening rate is 62.16% higher than the optimal result in the orthogonal test table, and the maximum thinning rate is in the optimal value area for the orthogonal test table. This optimization method provides some theoretical reference for hot stamping forming of 6016 aluminum alloys.

## Figures and Tables

**Figure 1 materials-15-08350-f001:**
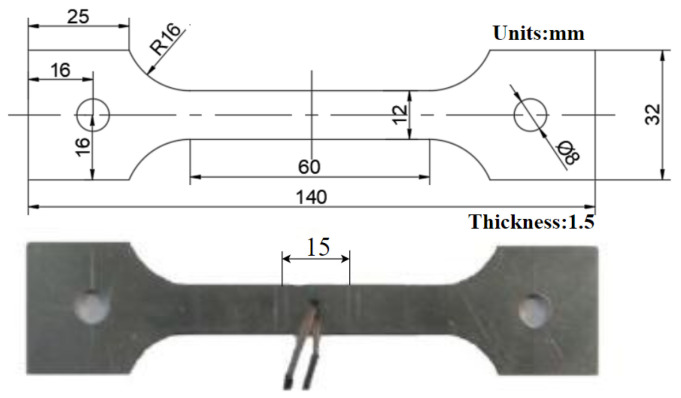
The shape and size of the sample.

**Figure 2 materials-15-08350-f002:**
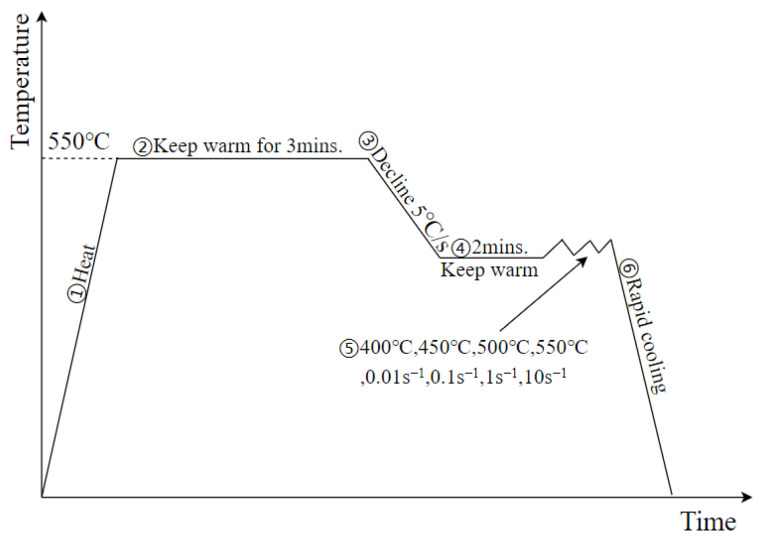
Temperature and deformation processes of the specimen.

**Figure 3 materials-15-08350-f003:**
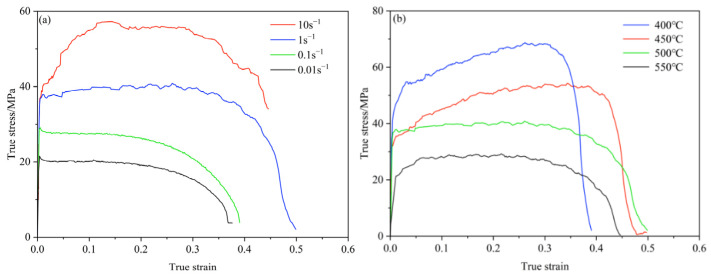
The stress–strain curve of 6016 aluminum alloy under different conditions. (**a**) The deformation temperature is 500 °C; (**b**) the strain rate is 1 s^−1^.

**Figure 4 materials-15-08350-f004:**
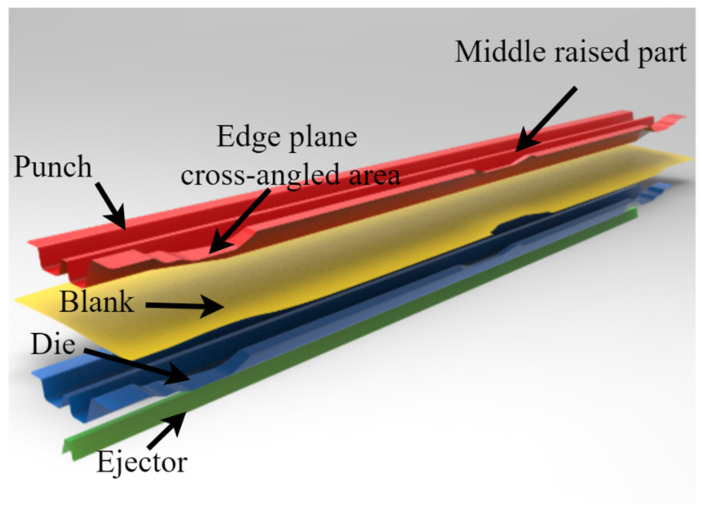
The finite element model for tests.

**Figure 5 materials-15-08350-f005:**
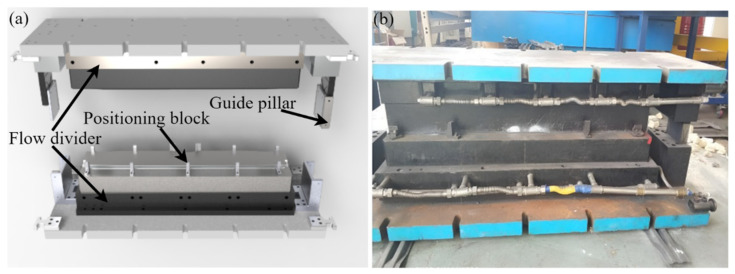
The toolset used for hot stamping tests. (**a**) 3D model. (**b**) Actual model.

**Figure 6 materials-15-08350-f006:**
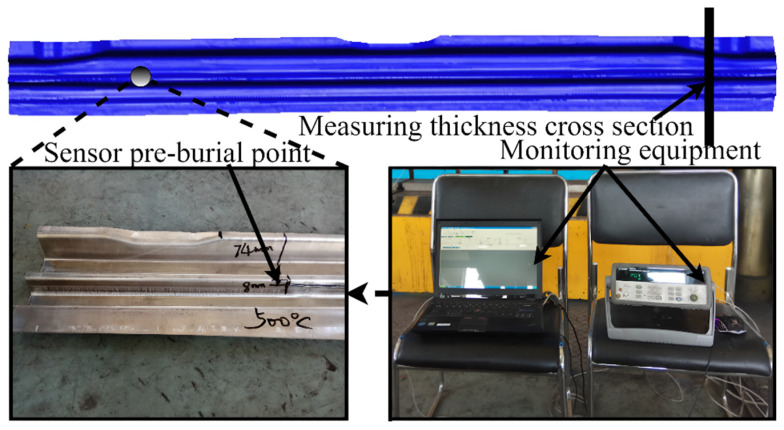
The experimental program.

**Figure 7 materials-15-08350-f007:**
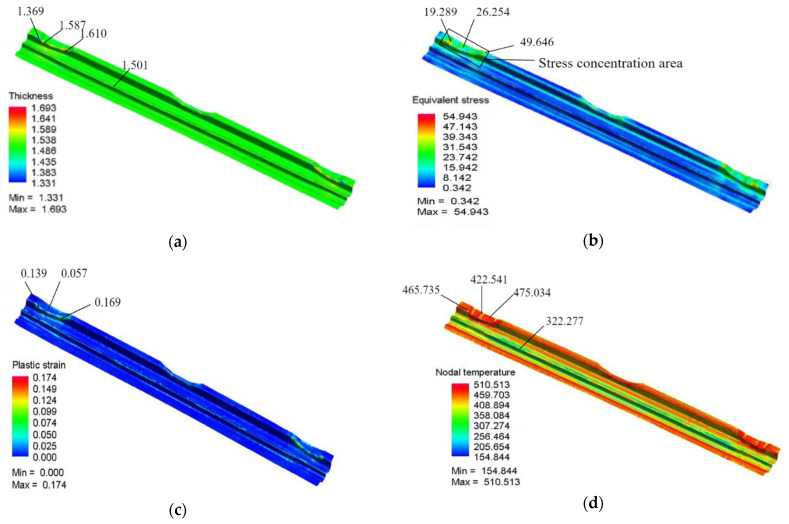
Distribution of simulation results: (**a**) Thickness distribution, (**b**) Stress distribution, (**c**) Strain distribution, and (**d**) Temperature distribution.

**Figure 8 materials-15-08350-f008:**
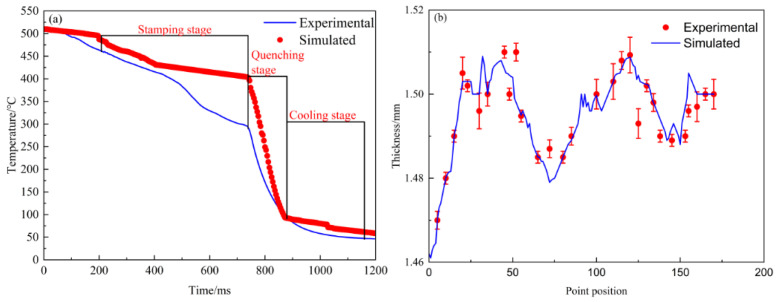
Comparison of experimental and simulated data: (**a**) Temperature variation, (**b**) Thickness distribution.

**Figure 9 materials-15-08350-f009:**
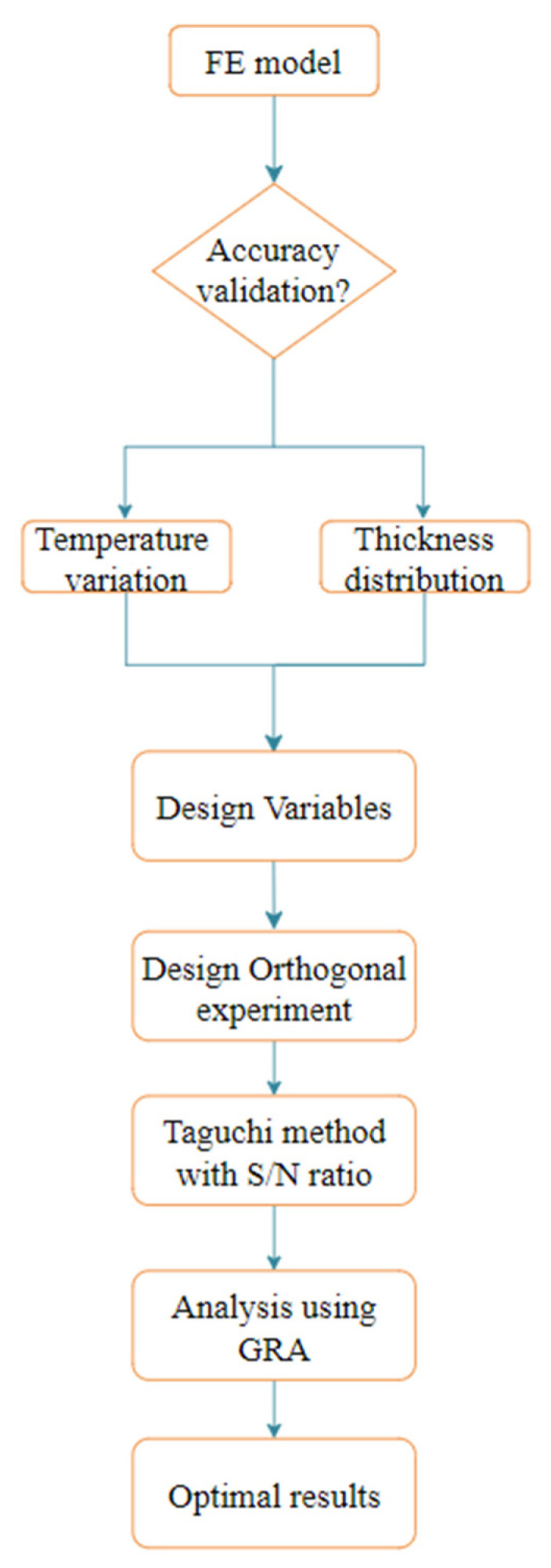
Flowchart of the optimal method.

**Figure 10 materials-15-08350-f010:**
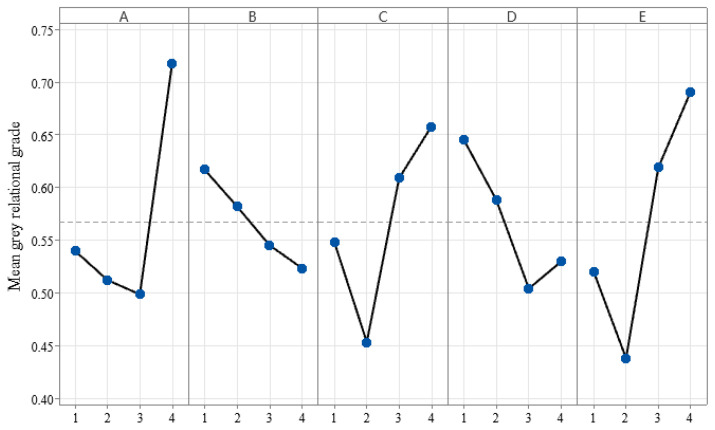
Main effects of factor level.

**Figure 11 materials-15-08350-f011:**
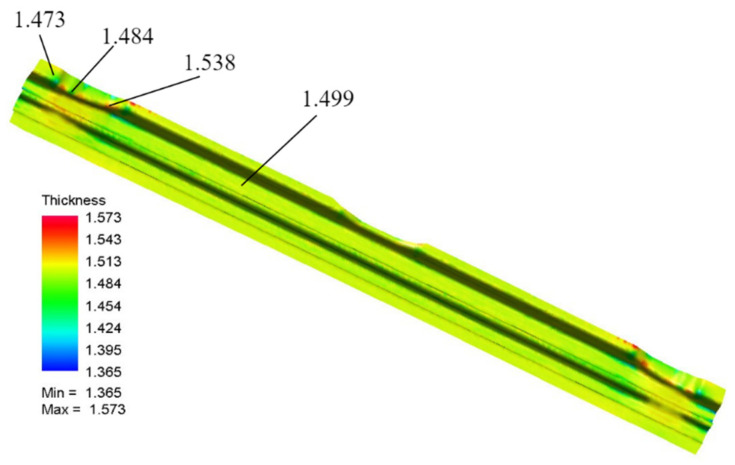
Optimization simulation results.

**Table 1 materials-15-08350-t001:** Chemical composition of 6016-T6 aluminum alloy (wt%).

Si	Fe	Cu	Mn	Mg	Cr	Zn	Ti	Al
1.0	0.18	0.14	0.08	0.44	0.05	<0.01	0.02	Balance

**Table 2 materials-15-08350-t002:** Thermal and physical properties of 6016 aluminum alloy.

Temperature (°C)	20	300	400	500
Thermal conductivity (W/(m·K))	119	130	139	141
Specific heat (J/(Kg·°C)	900	1040	1080	1100
Modulus of elasticity (MPa)	68,000	45,000	38,000	30,000

**Table 3 materials-15-08350-t003:** Heat transfer coefficients in FE analysis.

Pressure (MPa)	Gap (mm)	$\mathrm{Heat}\;\mathrm{Transfer}\;\mathrm{Coefficient}\;(W/m^{2}\cdot K)$
0	6	0
10	5	0.5
15	1	1.5
25	0	3

**Table 4 materials-15-08350-t004:** Factor level table.

	Blank Holder Force (kN)	Die Gap (mm)	Deformation Temperature (°C)	Coefficient of Friction	Stamping Speed (mm/s)
Level 1	20	1.05 t	480	0.15	50
Level 2	30	1.10 t	510	0.30	100
Level 3	40	1.15 t	540	0.45	150
Level 4	50	1.20 t	570	0.60	200

**Table 5 materials-15-08350-t005:** Orthogonal test scheme.

No.	Blank Holder Force (kN)	Die Gap (mm)	Deformation Temperature (°C)	Coefficient of Friction	Stamping Speed (mm/s)	Maximum Thinning Rate (%)	Maximum Thickening Rate (%)
1	20	1.05 t	480	0.15	50	19.00	12.13
2	20	1.10 t	510	0.30	100	21.47	34.87
3	20	1.15 t	540	0.45	150	13.67	15.73
4	20	1.20 t	570	0.60	200	11.00	13.40
5	30	1.05 t	510	0.45	200	14.73	16.80
6	30	1.10 t	480	0.60	150	10.47	26.73
7	30	1.15 t	570	0.15	100	14.07	16.20
8	30	1.20 t	540	0.30	50	15.33	17.93
9	40	1.20 t	540	0.60	100	14.27	27.13
10	40	1.05 t	570	0.45	50	11.73	24.00
11	40	1.10 t	480	0.30	200	9.80	19.87
12	40	1.15 t	510	0.15	150	19.47	16.40
13	50	1.05 t	570	0.30	150	8.07	11.93
14	50	1.10 t	540	0.15	200	8.27	10.87
15	50	1.15 t	510	0.60	50	14.67	17.60
16	50	1.20 t	480	0.45	100	16.33	18.67

**Table 6 materials-15-08350-t006:** Value of Grey Relational Coefficient and Grey Relational Grade.

No.	S/*n*		Normalized		GRC		GRG	Rank
	$\eta_{\mathrm{Thin}}$	$\eta_{\mathrm{Thick}}$	$X_{\mathrm{Thin}}$	$X_{\mathrm{Thick}}$	$\xi_{\mathrm{Thin}}$	$\xi_{\mathrm{Thick}}$		
1	−12.79	−10.83	0.13	0.90	0.363	0.841	0.602	4
2	−13.32	−15.42	0.00	0.00	0.334	0.334	0.334	16
3	−11.36	−11.97	0.46	0.68	0.481	0.617	0.549	6
4	−10.41	−11.27	0.68	0.82	0.612	0.736	0.674	3
5	−11.68	−12.25	0.38	0.63	0.448	0.572	0.510	8
6	−10.20	−14.27	0.73	0.23	0.653	0.393	0.523	10
7	−11.48	−12.09	0.43	0.66	0.468	0.593	0.531	7
8	−11.85	−12.54	0.34	0.57	0.432	0.538	0.485	11
9	−11.54	−14.33	0.42	0.21	0.462	0.389	0.425	15
10	−10.69	−13.80	0.62	0.32	0.566	0.424	0.495	13
11	−9.91	−12.98	0.80	0.48	0.715	0.491	0.603	5
12	−12.89	−12.15	0.10	0.65	0.357	0.586	0.472	12
13	−9.07	−10.77	1.00	0.92	1.000	0.862	0.931	2
14	−9.17	−10.36	0.97	1.00	0.952	1.000	0.976	1
15	−11.66	−12.45	0.39	0.59	0.450	0.547	0.498	9
16	−12.13	−12.71	0.28	0.54	0.409	0.518	0.463	14

**Table 7 materials-15-08350-t007:** Different level GRG Mean table (ANOR).

	Blank Holder Force (kN)	Die Gap (mm)	Deformation Temperature (°C)	Coefficient of Friction	Stamping Speed (mm/s)
	A	B	C	D	E
Level 1	0.571	0.635	0.544	0.678	0.543
Level 2	0.511	0.549	0.474	0.570	0.444
Level 3	0.480	0.543	0.606	0.510	0.611
Level 4	0.712	0.548	0.650	0.516	0.677
Delta	0.232	0.091	0.176	0.168	0.233
Rank	2	5	3	4	1

**Table 8 materials-15-08350-t008:** ANOVA for the GRG table.

Source	DOF	SS	MS	F	Contribution (%)
A	3	0.12359	0.041198	6.07	28.64
C	3	0.09293	0.030975	4.56	21.53
D	3	0.04752	0.015842	2.33	11.01
E	3	0.14711	0.049037	7.23	34.09
Error	3	0.02036	0.006787		4.72
Total	15	0.43151			100

## Data Availability

Data are contained within the article. The data presented in this study are available in Optimization of Process Parameter in 6016 Aluminum Alloy Hot Stamping Using Taguchi-grey Relational Analysis.

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
