# Peer review of "Multi-Objective Optimization of Process Parameters in 6016 Aluminum Alloy Hot Stamping Using Taguchi-Grey Relational Analysis"

_materials, 2022, doi:10.3390/ma15238350_

Round 1

Reviewer 1 Report

This manuscript mainly uses the simulation model, Taguchi design, GRA, ANOVA for multi-objective optimization of the aluminum alloy hot Stamping process, which has process application value. However, the following comments need to be supplemented and clarified before they can be considered for publication.

1.      Similar studies have been published in the literature, e.g. references 14-16. The authors should explain the differences and innovations between this study and these multi-objective optimization of hot stamping process papers to reinforce the motivation for the study.

2.      For simulation verification in Section 3.1, it is recommended to verify with the quality indicators concerned by the process: the maximum thickening rate and maximum thinning rate. In addition, the simulation model should also be checked for mesh independence.

3.      On page 7, it isn’t explain whether the maximum thickening rate and maximum thinning rate are “the larger the better” or “the smaller the better”. It should be clearly stated whether to use Equation (1) or Equation (2). Because these two indicators are very important, It is recommended that the manuscript should explain why these two indicators are used and how they are calculated, rather than just listing references.

4.      On Page 11, the maximum thickening rate on line 288 is 4.87%, but on line 291 is 62.16%, please confirm which is correct.

5. The optimization results were verified by the simulation results, can they be verified by the experimental results?

Reviewer 2 Report

The article in question concerns the manufacturing processes of elements made of aluminium alloys and their optimization which is a crucial topic in modern research and development works concerning production of construction elements. The authors achieved a high compliance between the finite element method calculations and experimental results. The obtained results may be useful in terms of optimization of hot stamping forming processes of aluminium alloys.

 Comments and recommendation regarding the suitable changes:

Table 1 presents the chemical composition of aluminium alloy 6016. It should be specified whether the tested samples had the exact chemical composition specified in the table or whether it is the chemical composition of the aluminium alloy 6016 required by the international standard.

 Fig. 1 shows the initial microstructure of aluminum alloy 6016 and it is a pity that it does not show the state after the heat treatment. The structure presented in Figure 1 does not contribute to the article by itself.

 Regarding Fig. 8, it is recommended to place the markers with numerical values - especially in the stress concentration area (due to the scales it is difficult to determine the values of stress, plastic strain, thickness and temperature). For comparative purposes, a similar recommendation applies to Fig. 12

Reviewer 3 Report

This manuscript reports on Multi-Objective Optimization of Process Parameters in 6016 Aluminum Alloy Hot Stamping Using Taguchi-grey Relational analysis. Authors should improve abstract and discussion part. I recommend a major revision.

1.     Please rewrite the abstract and add parameters range used to improve hot stamping.

2.     In introduction, please add why this specific alloy was used and describe previous attempts to optimize hot stamping of this alloy.

For example, following papers could be used

https://doi.org/10.1007/s11771-019-4024-8 

https://doi.org/10.4028/www.scientific.net/MSF.854.133 

Please provide an extensive literature search on this issue

3.     Please improve quality of Figures 5, 6 and 7 since the quality is insufficient. They should be clear visible in A4 format. Please use black colour instead of red in the Figures.

4.     Results are well presented, but more discussion is required

Round 2

Reviewer 1 Report

The manuscript can be accepted for publication

Author Response

Thank you for your reviewing.

Reviewer 3 Report

Paper could be accepted for publication

Author Response

Thank you for your reviewing